# Novel Electrochemical Aptasensor Based on Ordered Mesoporous Carbon/2D Ti_3_C_2_ MXene as Nanocarrier for Simultaneous Detection of Aminoglycoside Antibiotics in Milk

**DOI:** 10.3390/bios12080626

**Published:** 2022-08-10

**Authors:** Fengling Yue, Mengyue Liu, Mengyuan Bai, Mengjiao Hu, Falan Li, Yemin Guo, Igor Vrublevsky, Xia Sun

**Affiliations:** 1School of Agricultural Engineering and Food Science, Shandong University of Technology, Zibo 255049, China; 2Shandong Provincial Engineering Research Center of Vegetable Safety and Quality Traceability, Zibo 255049, China; 3Zibo City Key Laboratory of Agricultural Product Safety Traceability, Zibo 255049, China; 4Department of Information Security, Belarusian State University of Informatics and Radioelectronics, 220013 Minsk, Belarus

**Keywords:** broad-spectrum aptamer, aminoglycoside antibiotics, Ti_3_C_2_ MXene, ordered mesoporous carbon, electrochemical aptasensor

## Abstract

Herein, a novel electrochemical aptasensor using a broad-spectrum aptamer as a biorecognition element was constructed based on a screen-printed carbon electrode (SPCE) for simultaneous detection of aminoglycoside antibiotics (AAs). The ordered mesoporous carbon (OMC) was firstly modified on 2D Ti_3_C_2_ MXene. The addition of OMC not only effectively improved the stability of the aptasensor, but also prevented the stacking of Ti_3_C_2_ sheets, which formed a good current passage for signal amplification. The prepared OMC@Ti_3_C_2_ MXene functioned as a nanocarrier to accommodate considerable aptamers. In the presence of AAs, the transport of electron charge on SPCE surface was influenced by the bio-chemical reactions of the aptamer and AAs, generating a significant decline in the differential pulse voltammetry (DPV) signals. The proposed aptasensor presented a wide linear range and the detection limit was 3.51 nM. Moreover, the aptasensor, with satisfactory stability, reproducibility and specificity, was successfully employed to detect the multi-residuals of AAs in milk. This work provided a novel strategy for monitoring AAs in milk.

## 1. Introduction

As an important kind of antibiotic, aminoglycoside antibiotics (AAs) are widely applied in human therapy and animal husbandry because of their effective antibacterial activity and low cost [1,2,3]. AAs include gentamicin (GEN), neomycin (NEO), streptomycin (STR), kanamycin (KAN) and so on, which possess the common structure of a streptamine ring [4]. Nevertheless, the AAs multi-residues in foods may cause severe toxic side effects with ototoxicity and nephrotoxicity [5,6,7]. On account of the potential hazardous risks of AAs residues, the European Union (EU), China, America and so on have pointed out the maximum residue limits (MRLs) of AAs in foods, for example, setting MRLs for 200 μg/kg in a milk sample [8]. Even so, it has been reported that the multi-residues of AAs are found to exist in food products. If the class of antibiotics is able to be detected simultaneously, the analysis of a large number of samples will be avoided and the detection procedure will be simplified so as to improve the detection efficiency.

Various detection methods have been established and are currently used for simultaneous quantification of AAs, such as liquid chromatography-tandem mass spectrometry (LC-MS/MS) [9,10,11], enzyme-linked immunosorbent assay (ELISA) [12] and capillary electrophoresis (CE) [13]. However, their inherent shortcomings of complicated sample pretreatment, costly instruments and requirement for professional operation skill limit the on-site application of these methods. Thus, developing rapid and sensitive as well as convenient methods in simultaneous analysis of AAs is still required. For this purpose, aptasensor, as a type of biosensor with great promise, can achieve simultaneous detection of AAs with a broad-spectrum aptamer as the biorecognition element. Aptamers are short single-stranded DNA (ssDNA) or RNA sequences selected by systematic evolution of ligands by exponential enrichment (SELEX) [14,15]. Aptamers possess some prominent advantages over antibodies, such as thermal stability, easy to synthesize and modify, lack of immunogenicity and screened with no requirement for laboratory animals [16,17,18]. In our previous work, broad-spectrum ssDNA aptamers for AAs were successfully obtained through graphene oxide (GO)-SELEX, which possesses broad specificity for the class of AAs [19]. Thus, it can function as a recognition element in the fabrication of an aptasensor for simultaneously detecting AAs.

Notably, electrochemical aptasensors have aroused extensive attention on account of their unique properties of high sensitivity, multiplexed analysis, straightforward operation and rapid response time [20,21,22]. Electrochemical aptasensors can be constructed for quantitative determination of AAs based on the detectable electrical response signal produced by the bio-chemical reactions from the broad-spectrum ssDNA aptamer and targets on the surface of the electrode [23]. However, the detection signal change merely caused by the interactions of aptamer and target cannot afford to attain the requirement of sensitivity. Various nanomaterials are applied for amplifying response signal and enhancing detection performance in development of the aptasensor [24,25,26]. The ordered mesoporous carbon (OMC) was utilized in the fabrication of the aptasensor for simultaneous detection of AAs in our previous research and the aptasensor displayed a linear range, from 10 nM to 1000 nM, with limit of detection (LOD) of 2.47 nM [19]. However, due to the loose porous structure and poor film-forming ability of OMC, the OMC was prone to drop from the electrode surface during detection. Then, OMC could not be well modified on the electrode surface. Two-dimensional Ti_3_C_2_ MXene, as a new type of nanomaterial, exhibits the unique features of high electrical conductivity, large specific surface with abundant active groups and desirable biocompatibility [27]. Due to these superior properties, Ti_3_C_2_ MXene has been extensively applied in electrochemical aptasensors [28,29]. However, in the construction process of an aptasensor, the key problem for Ti_3_C_2_ MXene is the aggregation occurrence, which may reduce the specific surface area and affect the stability of MXenes. Hence, we attempt to modify OMC on Ti_3_C_2_ MXene and then prepare nanocomposites of OMC@Ti_3_C_2_ MXene. In this case, it can take full advantage of both OMC and Ti_3_C_2_ MXene. The OMC@Ti_3_C_2_ MXene nanocomposites could be well modified on the electrode surface because of the excellent layer structure of Ti_3_C_2_ MXene. Moreover, the addition of OMC not only effectively improves the stability of the aptasensor, but also prevents the stacking of Ti_3_C_2_ sheets and, remarkably, enhances the detection performance of the aptasensor because of the unique mesostructure of OMC.

Herein, a novel electrochemical aptasensor using the broad-spectrum aptamer as a biorecognition probe was constructed based on screen-printed carbon electrode (SPCE) modification with OMC@Ti_3_C_2_ MXene for simultaneous analysis of AAs in milk. On one hand, the OMC@Ti_3_C_2_ MXene could promote electron transfer on the SPCE surface to amplify the signal of the aptasensor. On the other hand, OMC@Ti_3_C_2_ MXene could be utilized as a nanocarrier for accommodating a large number of aptamers for recognizing and capturing targets. The morphology properties and component analysis of OMC@Ti_3_C_2_ nanocomposites were characterized. Furthermore, cyclic voltammetry (CV) and electrochemical impedance spectroscopy (EIS) were used for analyzing the stability and conductivity of OMC@Ti_3_C_2_ nanocomposites. The change in transfer electrons on the electrode surface between before and after aptamer incubation with AAs could be expressed through differential pulse voltammetry (DPV) signals for quantitative analysis of AAs. The designed aptasensor exhibited a wide linear range and low detection limit, with high sensitivity and good specificity. Furthermore, it was successfully applied in simultaneous determination of AAs in spiked milk samples.

## 2. Materials and Methods

### 2.1. Reagents and Materials

The broad-spectrum aptamer for AAs was synthesized and purified from Sangon Biotech Co., Ltd. (Shanghai, China) and the sequence was as follows: 5′-CGGATCCCCAGCTCGGGGTGCTATGGAGGCTGTATCGGAGACCTGCAGG-3′. AAs including neomycin (NEO), kanamycin (KAN), gentamicin (GEN), streptomycin (STR), dihydrostreptomycin (DH-STR), tobramycin (TOB), spectinomycin (SPE), apramycin (APR), paromomycin (PAR), amikacin (AMI) and other antibiotics including tetracycline (TET), sulfadiazine (SUL), erythromycin (ERY), chloramphenicol (CHL) and ampicillin (AMP) were bought from Macklin Biochemical Co., Ltd. (Shanghai, China). Dimethyl sulfoxide (DMSO) and chitosan (CS) were purchased from Aladdin industrial Co. (Shanghai, China). Bovine serum albumin (BSA) was obtained from BioDev-Tech. Co. (Beijing, China). Ti_3_AlC_2_ was got from XFNANO Materials Tech Co., Ltd. (Nanjing, China). OMC was purchased from Yoshikura Nanotechnology Co. (Nanjing, China). The [Fe(CN)_6_]^3−/4−^ solution containing 5.0 mM K_4_[Fe(CN)_6_], 5.0 mM K_3_[Fe( CN)_6_] and 0.1 M KCl was prepared with ultrapure water as solvent. All chemicals were of analytical grade. All aqueous solutions in the experiment were prepared with ultrapure water obtained by water purification system (PALL, New York, NY, USA, 18.2 MΩ·cm at 25 °C).

### 2.2. Apparatus

SPCE was assembled with three-electrode system with a carbon working electrode, a carbon counter electrode and a Ag/AgCl reference electrode, which was bought from Zensor R&D (Taiwan, China). The electrochemical measurements were tested on a CHI660D electrochemical workstation (Shanghai Chenhua Instruments Co., Shanghai, China). Scanning electron microscope (SEM) images were acquired on an FEI Sirion 200 scanning electron microscope (FEI, Hillsboro, OR, USA). Transmission electron microscopy (TEM) images were collected with an FEI Tecnai G2F20S-TWIN transmission electron microscopy (FEI, Hillsboro, OR, USA). Atomic force microscopy (AFM) images were produced by an SPM-9700 atomic force microscope (Shimadzu, Tokyo, Japan). X-ray photoelectron spectroscopy (XPS) measurements were carried out on ESCALAB 250Xi electron spectrometer (Thermo Fisher Scientific, Co., Waltham, MA, USA.). The quantitative analysis of AAs in milk samples was carried out on the Agilent 6470 LC-MS/MS equipped with triple quadrupole MS (Agilent Technologies, Santa Clara, CA, USA). Electrospray positive ionization (ESI+) was used for ionization of the target analytes. Multiple reaction monitoring (MRM) mode was used to determine the quantitative and qualitative ions and the transitions of the compounds are shown in Table 1. A C18 column was used for chromatographic separation. The elution program was gradient elution with methanol(A) and 0.1% formic acid aqueous solution (B) including 0 min A10%, B90%; 1 min A10%, B90%; 2 min A80%, B20%; 5 min A80%, B20%; and 6 min A10%, B90%.

### 2.3. Preparation of Ti_3_C_2_ MXene 

The Ti_3_C_2_ Mxene was prepared referring to the previous literature [30]. Firstly, addition of 1.0 g of Ti_3_AlC_2_ powders into 10 mL of hydrofluoric acid (HF) step by step was performed and the mixtures were stirred at 45 °C for 24 h. Subsequently, the mixtures were centrifuged so as to obtain sediment. After rinsing the sediment by ultrapure water, the final pH of the solution was 6 and then it was left to dry naturally. Last, the obtained product was immersed in 1 mL of DMSO with stirring for 24 h at room temperature. The suspension was rinsed by ultrapure water, therewith centrifugation at 3500 rpm for 1 h. The collected black precipitate was dried at 60 °C and then stored at 4 °C before use. 

### 2.4. Preparation of OMC@Ti_3_C_2_ MXene 

Further, 0.1 g of chitosan powder was dispersed in 1.0% acetic acid with stirring for 10 h to obtain 0.2% CS solution. Afterwards, 5 mg of OMC powder was dissolved completely into 5 mL of 0.2% CS solution with ultrasonic treatment so as to acquire a homogeneous suspension of OMC. Subsequently, 5 mL of the OMC homogeneous suspension (1 mg/mL) was ultrasonically dispersed in 5 mL of Ti_3_C_2_ Mxene (0.1 mg/mL). The obtained OMC@Ti_3_C_2_ MXene composite materials were stored at 4 °C for further experiments.

### 2.5. Fabrication of the Aptasensor

Before assembling the aptasensor, SPCE was pretreated with CV scanning in 0.5 M H_2_SO_4_ for 25 cycles from −1.5 V to 1.5 V [31]. Then the SPCE surface was cleaned by ultrapure water for next use. Figure 1 shows the fabrication process of aptasensor and the measurements were carried out on electrochemical station. Firstly, 8 μL of OMC@Ti_3_C_2_ MXene was used to modify the surface of SPCE. After drying, the modified SPCE was covered with 8 μL of aptamer solution (5 μM) and incubated at room temperature. Afterwards, the SPCE sensing surface were incubated with 8 μL of 0.5% BSA solution for 30 min so as to block the nonspecific sites, then obtaining the fabricated aptasensor. For the purpose of determination of AAs, the aptasensor was incubated with 8 μL of equimolar mixture of targets containing GEN, NEO, KAN, AMI, PAR, APR, SPE, TOB, DH-STR and STR with final concentrations of 10, 50, 100, 250, 500, 750, 1000, 1500 and 2000 nM, then left to dry in the air.

The concentration of AAs was determined via measuring the change in DPV current response. DPV measurement was conducted in [Fe(CN)_6_]^3−/4−^ solution ranging from −0.1 V to 0.5 V with a pulse time of 0.2 s and pulse amplitude of 50 mV. As shown in Figure 1b, on the surface of SPCE, the aptamer could fold a certain spatial structure and provide binding pockets for AAs, and then selectively capture AAs to be capped in binding pockets, forming the complex, thus, impeding the electron transfer and generating signal reduction. In this way, the change in transfer electrons on the surface of electrode could be expressed by DPV current. Then, the value of ΔI based on the DPV current between before and after AAs incubation was measured for detection of AAs.

### 2.6. Pretreatment of Milk Samples

The applicability of the proposed aptasensor was evaluated in real milk samples. Firstly, sample solutions with final AAs concentrations of 0, 50 nM, 100 nM, 500 nM and 1000 nM were prepared on the basis of spiked milk samples. Secondly, the solutions were centrifuged for 20 min at 10,000 rpm so as to remove the upper fat. Subsequently, the solutions were filtrated by 0.22 μm sterile Millipore membrane. Finally, 10-fold dilution of the obtained sample solutions with PBS solution (pH 7.5, 0.01 M) was utilized for further sample experiments.

## 3. Results and Discussion 

### 3.1. Characterization of Nanomaterials

The morphology of Ti_3_C_2_ MXene was characterized by TEM. From Figure 1A, the prepared Ti_3_C_2_ MXene emerged in sheets and possessed large specific surface area for enhancing electron charge transfer and improving electrical conductivity. However, the single-component Ti_3_C_2_ nanosheets could be easily stacked together, owing to a strong hydrogen bond and Van der Waals interaction between adjacent Ti_3_C_2_ nanosheets. The re-stacking of Ti_3_C_2_ nanosheets seriously hindered the ion transport channel, further affected and limited the actual electrochemical performance of Ti_3_C_2_ MXene [32]. The TEM image of OMC is shown in Figure 1B and it was observed that the structure of OMC presented loose and porous morphology. The shapes of the pores were diverse. Then, it could form a better current passage for producing high conductivity. We modified the OMC on the surface of Ti_3_C_2_ MXene and then prepared OMC@Ti_3_C_2_ MXene nanocomposites. The surface morphology of OMC@Ti_3_C_2_ MXene is characterized by SEM in Figure 1C. It showed the porous OMC could embed into the Ti_3_C_2_ nanosheets, which avoided the re-stacking of Ti_3_C_2_ nanosheets. The surface roughness of Ti_3_C_2_ MXene and OMC@Ti_3_C_2_ was further characterized by AFM. The surface topography of Ti_3_C_2_ MXene showed relative smoothness and the surface roughness was 1.195 nm (Figure 1D). When the OMC was modified on Ti_3_C_2_ MXene, the surface topography of OMC@Ti_3_C_2_ became more rugged and the surface roughness increased to 3.331 nm (Figure 1E). Then, it showed the surface area of nanocomposites increased, which could well act as a nanocarrier for accommodating numerous aptamers.

The elemental analysis of Ti_3_C_2_ MXene and OMC@Ti_3_C_2_ was studied through XPS patterns. It was observed that the characteristic peaks of Ti, C, O and F elements appeared in the survey spectrum of the Ti_3_C_2_ MXene and OMC@Ti_3_C_2_ phase (Figure 1F). From Figure 1G, the Ti 2p high-resolution spectra of Ti_3_C_2_ MXene showed two peaks at 461.2 eV and 454.9 eV, which were derived from Ti–C bands in the main structure. The other peaks appeared at 455.8 eV, 456.7 eV, 458.8 eV and 462.8 eV, which were assigned to Ti−O bond of Ti_3_C_2_ MXene surface. Further, as shown in Figure 1H, the characteristic peaks of Ti–C in the OMC@ Ti_3_C_2_ MXene changed to become 461.9 eV and 455.4 eV and the peaks of Ti−O bond in the OMC@ Ti_3_C_2_ MXene appeared at 456.1 eV, 456.8 eV, 459.3 eV, 463.8 and 465.0 eV. The Ti–C and Ti−O bond of OMC@ Ti_3_C_2_ MXene were distinct from Ti_3_C_2_ MXene, indicating the successful preparation of OMC@ Ti_3_C_2_ MXene.

The effect of Ti_3_C_2_ MXene and OMC@Ti_3_C_2_ MXene on the stability of the current response of the electrode was investigated. As shown in Figure 2A, when the electrode surface was modified with Ti_3_C_2_ MXene, the current response sharply decreased with the increase in the cycle times, which was attributed to the easy oxidation of Ti_3_C_2_ MXene. After four cycles, the current response of the electrode reached a very low level and severely affected the application of the aptasensor. As modification of the OMC on the Ti_3_C_2_ MXene surface, the current response signal of the electrode slightly increased and tended to be stable as the cycle times increased. In comparison with Ti_3_C_2_ MXene, the OMC@Ti_3_C_2_ MXene nanocomposites could significantly improve the stability of the electrode, for the reason that when the OMC was modified on the surface of Ti_3_C_2_ MXene, it could act as a stabilizer and prevent the oxidation of Ti_3_C_2_ MXene through reducing its exposure to oxygen in the air.

CV and EIS were measured to investigate the electrochemical behavior of Ti_3_C_2_ MXene, OMC and OMC@Ti_3_C_2_ MXene on the electrode surface (Figure 2B,C). From Figure 2B, the CV curve of the electrode with [Fe(CN)_6_]^3−/4−^ as electroactive probes had a pair of apparent reversible redox peaks. When the SPCE surface was modified by Ti_3_C_2_ MXene, the reductive peak current obviously became strong (curve d) compared with the bare electrode (curve c), which might result from the strong reduction ability of Ti_3_C_2_ MXene. Meanwhile, on account of the unique mesostructure of OMC, after OMC was dropped onto the electrode surface, an increased electrochemical signal could be obtained by CV (curve b). Furthermore, the immobilization of OMC@Ti_3_C_2_ MXene caused the peak current to increase remarkably (curve a) and the electrode reached a superior level. It was due to OMC embedding into the Ti_3_C_2_ nanosheets, which produced good current passage and further enhanced conductivity in the electrode. As well, EIS measurements were conducted in [Fe(CN)_6_]^3−/4−^ to further confirm the electrochemical properties of nanomaterials. In the EIS, the semicircle portion was the high-frequency region, showing the control process of electron charge transport on the modified electrode surface. Its diameter represented the ability of electron transfer impedance (Ret) [33]. In Figure 2C, it was observed that the Ret in the bare electrode was bigger (curve c). When the electrode surface was modified by the functional nanomaterials, the Ret value further decreased (curve a, b, d), suggesting that the nanomaterials may be able to accelerate the process of electron transfer in [Fe(CN)_6_]^3−/4−^ for improving the conductivity in the electrode. All of these EIS consequences were inconsistent with the CV characterization, indicating that the prepared nanomaterials had good electrochemical behavior. Thus, modification of OMC@Ti_3_C_2_ MXene on the electrode surface not only improved the stability of the designed aptasensor, but also enforced the conductivity in the electrode.

### 3.2. Electrochemical Characterization of Assembly Fabrication of the Aptasensor

CV, as the effective method, was applied to demonstrate the electrochemical characterization of the step-by-step assembly process of the aptasensor. As illustrated in Figure 3A, when the prepared OMC@Ti_3_C_2_ MXene was assembled on the SPCE sensing surface, the peak current increased obviously (curve a) in comparison with the current response of the bare electrode (curve e). The OMC@Ti_3_C_2_ MXene was dissolved in the chitosan solution, which possessed beneficial biocompatibility, enabling the aptamer to be well fixed on the surface of the electrode. Under this condition, OMC@Ti_3_C_2_ MXene provided large specific surface to load more aptamers. Then, the aptamer was coated on the modified SPCE, generating a decline in the peak current (curve b). It was probably because the negative-charged groups of aptamers restricted the mass transport of the redox probe, causing the signal to decrease. Further, the electron transfer was further hindered after treatment of the electrode with BSA for blocking the unbound sites (curve c), as expected because of the non-electroactive characteristics of BSA. Finally, in the presence of AAs, the aptamer had the capacity of selective binding to AAs and formed a stable complex on the SPCE surface, which restrained the transport of electron charge and caused a reduction in the peak current (curve d).

In addition, the feasibility of the aptasensor was characterized by testing the DPV signal in the [Fe(CN)_6_]^3−/4−^ (Figure 3B). The DPV current enhanced remarkably after modification of OMC@Ti_3_C_2_ MXene for inducing signal amplification (curve a). Afterwards, addition of aptamer generated the DPV current decrease (curve b) due to the biomolecule impeding the electron transfer. Similarly, the participation of BSA made the DPV current reduce again (curve e). Finally, the constructed aptasensor was incubated with AAs, which generated a further decline in the DPV current (curve d). The change in DPV current relied on the amount of target AAs. The concentration of AAs could be determined according to the corresponding value of ΔI between before and after AAs incubation. The prepared aptasensor was proved to be successfully fabricated on the basis of these results.

### 3.3. Optimization of Experimental Conditions

To achieve the superior sensing performance of the designed aptasensor for AAs detection, the crucial factors that influenced the current response were taken into account to optimize, including the aptamer concentration, the pH value and the incubation time, as well as the immobilization time for the aptamer.

As illustrated in Figure 4A, with increasing the concentration of aptamer, the value of ΔI attained the maximum value at an aptamer concentration of 5 μM and then remained at a stable level, signifying the amount of aptamer was sufficient for capturing targets on the SPCE surface. In view of the nonspecific adsorption of other pollutants caused by excessive aptamer, the concentration of aptamer as 5 µM was selected in subsequent experiments. In addition, the pH value of [Fe(CN)_6_]^3−/4−^ might affect the binding ability of aptamer and AAs and was also optimized in Figure 4B. When the pH value was 7.5, it was suitable for the combination of aptamer and AAs. The activity of the aptamer might be damaged in high acidity or alkalinity; in this case, the response of the aptasensor decreased. Thus, an optimal pH value of 7.5 was chosen. In addition, the incubation time between aptamer and AAs was investigated in Figure 4C. The incubation time was controlled by adding a special cover to prevent the volatilization of the solution. After incubation, the electrode was dried in the air and then DPV current of the electrode was measured. When the incubation time exceeded 40 min, the current signals of the aptasensor almost remained stable, implying the aptamer fully bound to AAs. Then, incubation time of 40 min was used for the following electrochemical analysis. In addition, the influence of immobilization time for aptamer on the sensing response of aptasensor was further investigated. From Figure 4D, it could be observed that 60 min was suitable for a well-immobilizing aptamer on the electrode surface for bio-recognition reaction. Then, the immobilization time for the aptamer was set as 60 min.

### 3.4. Electrochemical Analysis of the Aptasensor

Under the abovementioned optimal parameters, the DPV response of the fabricated aptasensor was investigated by incubation with solutions containing different AAs concentrations. The DPV current gradually decreased with the increase in AAs concentration in Figure 5 (curve a–j). This is because aptamers bound to more AAs, which formed the more complex on the SPCE surface, ultimately leading to a decline in the DPV response. Figure 5 shows the linear relationship of the logarithm of AAs concentrations (10–2000 nM) and ΔI value. The calibration curve was fitted as ΔI = −7.450 + 8.566LgC_AAs_ with a coefficient (R^2^) of 0.988 (*n* = 5). The LOD was calculated as 3.51 nM (S/N = 3). Therefore, the constructed aptasensor possessed good detection performance with wide detection range and low limit of detection.

### 3.5. Specificity, Stability and Reproducibility of the Aptasensor

The specificity of the aptaseneor for AAs was studied by testing AAs and other interfering antibiotics. As illustrated in Figure 6A, when there were no antibiotics in the blank samples, a little change in the DPV current was observed. The change in the DPV response of AAs containing KAN, NEO, TOB, DH-STR, GEN, SPE, STR, AMI, APR and PAR had evident increases, which indicated the aptasensor possessed acceptable detection performance for the class of AAs. Meanwhile, obvious DPV current changes, likewise, were presented in mixtures containing AAs and other interfering antibiotics; thus, the co-existence of the interfering antibiotics did not influence the aptasensor’s detectability. However, the change in the current was extremely low in the presence of only interferences antibiotics, including CHL, PEN, EPY, TET and SUL, which showed the aptasensor with good specificity for the class of AAs.

To assess the stability of the designed aptasensor, we prepared twenty-eight modified SPCEs under the same condition. The four SPCEs were incubated with 500 nM AAs and then the DPV signal was tested. Afterwards, the remaining twenty-four SPCEs were stored at 4 °C and four SPCEs were taken out every 3 days for detection of AAs. As exhibited in Figure 6B, the DPV current of the aptasensor decreased by 6.42% with relative standard deviation (RSD) of 4.32% after 9 days of storage. Furthermore, the aptasensor retained 90.18% compared with the original DPV signals with RSD of 5.41% after 18 days of storage, which was more stable than our previous aptasensor [19]. Thus, it indicated the developed aptasensor possessed good stability. 

The reproducibility of the aptasensor was tested. Four identical aptasensors with preparation in the same case were used for measuring the same concentration of AAs at 1 × 10^3^ nM. The RSD of the four aptasensors was 3.10%, which demonstrated good reproducibility of this aptasensor (Figure 6C).

### 3.6. Analysis in Milk Samples

For the purpose of verifying the practicability of the developed aptasensor, it was applied in detection of AAs in real milk samples. Firstly, LC-MS/MS was employed for quantitative assay of the AAs content in milk samples. It was found that there were no AAs residues in these samples. Then, the spiked milk samples were prepared with different AAs concentrations of 0, 50, 100, 500 and 1000 nM. As demonstrated in Table 2, the recovery results were in a range of 97.01–106.90% with a satisfactory RSD from 1.59% to 4.53%, demonstrating the proposed aptasensor with acceptable practicability could be a promising analytical approach for accurate analysis of AAs in milk samples.

## 4. Conclusions

In this experiment, a novel electrochemical aptasensor with a broad-spectrum aptamer as the biorecognition element was constructed for simultaneous determination of AAs. The prepared OMC@Ti_3_C_2_ MXene was utilized to modify the electrode; in this way, it not only improved the stability of the aptasensor, but also provided large specific surface for acting as a nanocarrier so as to accommodate a large number of aptamers for recognizing and capturing targets. Based on the abovementioned advantages, the fabricated aptasensor had a wide linear range of 10–2000 nM and a low LOD of 3.51 nM. Meanwhile, the aptasensor exhibited desirable specificity, good stability and reproducibility. Moreover, it was successfully employed in simultaneously detecting AAs in the spiked milk samples. Therefore, this strategy could offer a useful tool for simultaneous analysis of antibiotic residue in the food safety field.

## Data Availability

Not applicable.

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
