# Peer review of "Novel Electrochemical Aptasensor Based on Ordered Mesoporous Carbon/2D Ti3C2 MXene as Nanocarrier for Simultaneous Detection of Aminoglycoside Antibiotics in Milk"

_biosensors, 2022, doi:10.3390/bios12080626_

Round 1

Reviewer 1 Report

The manuscript describes a novel electrochemical aptasensor using the broad-spectrum aptamer as biorecognition element based on ordered mesoporous carbon/2D Ti3C2 MXene as nanocarrier for simultaneous detection of aminoglycoside antibiotics in milk. This work provides a novel strategy for simultaneous detection of AAs in milk. It is meaningful and useful. The manuscript has done a lot of work and well organized. However, there are some problems in the article that need to be revised carefully. I recommend the publication of the manuscript in "Biosensors" after minor revision.

Specific points that should be improved:

1. The manuscript has some grammatical problems, such as "AAs include gentamicin (GEN), neomycin (NEO), streptomycin (STR), kanamycin (KAN) and so on, which possessed the common structure of streptamine ring" in the line 40-42; "It was observed in Figure 2C, the Ret of bare electrode is bigger (curve c)" in the line 249. …, check the whole manuscript for spelling mistakes and grammar.

2. In the line 163, why did Authors give Fe2+/Fe3+ to the sample solutions?

3. Figure 4 should be improved and there is no ordinate title in the Figure 4 (D).

4. The labels of the Figure 6 are not clear. The three graphs (Specificity, stability and reproducibility of aptasensor) in the Figure 6 should be separated.

5. There are some errors in the reference, such as line 399-400 " Dillard, L.K.; Wu, C.Z.; Saunders, J.E.; McMahon, C.M. A scoping review of global aminoglycoside antibiotic overuse: A poten- 399

tial opportunity for primary ototoxicity prevention. Res. Soc. Admin. Pharm. 2021 "; line 459-460" Zhang, H.; Wang, Z.; Zhang, Q.; Wang, F.; Liu, Y. Ti3C2 MXenes nanosheets catalyzed highly efficient electrogenerated chemiluminescence biosensor for the detection of exosomes. Biosens. Bioelectron. 2019, 124-125, 184-190. "…, check the reference and correct them.

Author Response

The manuscript describes a novel electrochemical aptasensor using the broad-spectrum aptamer as biorecognition element based on ordered mesoporous carbon/2D Ti3C2 MXene as nanocarrier for simultaneous detection of aminoglycoside antibiotics in milk. This work provides a novel strategy for simultaneous detection of AAs in milk. It is meaningful and useful. The manuscript has done a lot of work and well organized. However, there are some problems in the article that need to be revised carefully. I recommend the publication of the manuscript in "Biosensors" after minor revision.

Specific points that should be improved:

  1. The manuscript has some grammatical problems, such as "AAs include gentamicin (GEN), neomycin (NEO), streptomycin (STR), kanamycin (KAN) and so on, which possessed the common structure of streptamine ring" in the line 40-42; "It was observed in Figure 2C, the Ret of bare electrode is bigger (curve c)" in the line 249. …, check the whole manuscript for spelling mistakes and grammar.

Response: We agree with you and thanks for your suggestion. We have corrected these sentences as follows:

AAs include gentamicin (GEN), neomycin (NEO), streptomycin (STR), kanamycin (KAN) and so on, which possess the common structure of streptamine ring. (line 41-43)

In Figure 2C, it was observed that the Ret of bare electrode was bigger (curve c). (line 270-271)

We have carefully checked the manuscript from cover to cover and corrected these problems in sentences of this article. And the revised content is marked in our revised manuscript.

  1. In the line 163, why did Authors give Fe2+/Fe3+ to the sample solutions?

Response: Thank you for your question. The redox couple [Fe(CN)6]3-/4- was used as electroactive probes by the reaction of Fe2+/Fe3+ for monitoring the modifications in the electron charge transfer induced by the molecule adsorption. The CV curve of the modified SPCE in [Fe(CN)6]3-/4- had a pair of distinct reversible redox peaks. The ability of the electrode surface to transfer electrons of the [Fe(CN)6]3-/4- ions was expressed by DPV signal. In this way, CV and DPV curve was used to characterize the electrochemical behavior of different modified SPCE. In the presence of AAs, the aptamer could specifically recognize AAs and form the complex with AAs on SPCE, which impeded the transfer of electron and generated signal decrease. The concentration of AAs in sample solutions was determined by linear regression equation on the basis of DPV current signal in [Fe(CN)6]3-/4-.

  1. Figure 4 should be improved and there is no ordinate title in the Figure 4 (D).

Response: Thank you very much for reminding me. We are sorry for our mistakes. We have added the ordinate title in the Figure 4 (D) in our revised manuscript.

  1. The labels of the Figure 6 are not clear. The three graphs (Specificity, stability and reproducibility of aptasensor) in the Figure 6 should be separated.

Response: Thanks for your suggestion. We have adjusted the three graphs in the Figure 6 in our revised manuscript.

  1. There are some errors in the reference, such as line 399-400 " Dillard, L.K.; Wu, C.Z.; Saunders, J.E.; McMahon, C.M. A scoping review of global aminoglycoside antibiotic overuse: A potential opportunity for primary ototoxicity prevention. Res. Soc. Admin. Pharm. 2021 "; line 459-460" Zhang, H.; Wang, Z.; Zhang, Q.; Wang, F.; Liu, Y. Ti3C2 MXenes nanosheets catalyzed highly efficient electrogenerated chemiluminescence biosensor for the detection of exosomes. Biosens. Bioelectron. 2019, 124-125, 184-190. "…, check the reference and correct them.

Response: Thank you very much for reminding me. We have corrected the references as follows:

1. Dillard, L.K.; Wu, C.Z.; Saunders, J.E.; McMahon, C.M. A scoping review of global aminoglycoside antibiotic overuse: A potential opportunity for primary ototoxicity prevention. Res. Soc. Admin. Pharm. 2022, 18, 3220-3229. (line 428-429)

30. Zhang, H.; Wang, Z.; Zhang, Q.; Wang, F.; Liu, Y. Ti3C2 MXenes nanosheets catalyzed highly efficient electrogenerated chemiluminescence biosensor for the detection of exosomes. Biosens. Bioelectron. 2019, 124-125, 184-190. (line 488-489). The volumes of this reference in Biosens. Bioelectron is 124–125.

Reviewer 2 Report

The manuscript describes the development of a  based on MXene  for antibiotic detection. The manuscript is within the scope of the journal with good scientific significance. The only suggestion for authors is that the figures need better quality and resolution.

Author Response

The manuscript describes the development of a  based on MXene  for antibiotic detection. The manuscript is within the scope of the journal with good scientific significance. The only suggestion for authors is that the figures need better quality and resolution.

Response: Thank you very much for your comments on the manuscript. We have increased the figures resolution in the revised manuscripts.

Reviewer 3 Report

In the presented manuscript, a novel aptasensor based on ordered mesoporous carbon/2D Ti3C2 MXene was fully characterized and successfully utilized in real samples analysis. The authors already presented a similar aptasensor for AAs detection in milk (please see ref. 19). The previous aptasensor was based on ordered mesoporous carbon only; however, it exhibited a similar performance. A comparison of those two aptasensors should be made, the differences underlined and demonstrate how the introduction of Ti3C2 MXene influenced its performance. 

Furthermore, some discrepancies occur in the text, especially in the ‘Material and Methods’ section, which may confuse the reader. Also, some details on the experimental setup are missing.  

  1. The Authors stated that broad spectrum aptamer were synthesized. The plural form indicates that more than one sequence was tested.
  2. The SPCE electrode was utilized in the presented work. The type of counter and reference electrode should be specified as well.
  3. The liquid chromatography-tandem mass spectrometer was used for the quantitative analysis of AAs in milk samples. For clarity, it should be specified that triple quadrupole type MS was used. The ion source (ESI/ APCI/ AJS ?), operation mode (MRM mode could be used for quantitative analysis), column type, and elution program should also be indicated.
  4. In the description of the Ti3C2 MXene preparation, at the final step, the supernatant was collected after centrifugation and stored before use. It is rather bizarre as the MXene is the precipitate.
  5. The methodology for the calibration of the aptasensor is not clear. In the current state, it seems that the aptasensor was incubated in the mixture of 10 different AAs of a final concentration of 10, 55, or 100, etc. Was that an equimolar mixture? What was the sensor performance (sensitivity, LOD, LOQ, linear range) in the presence of only one antibiotic? 8 µL of the AAs solution was dropped on the aptasensor and left to dry. Then, the sensor was placed in Fe(CN)6 3-/4- solution, and the DVP signal was registered. How did this drying step influence the aptasensor response? Were any control experiments conducted? Various incubation times with AAs were evaluated. Was the drying step involved here as well?
  6. The DPV and EIS measurements were conducted using Fe(CN)6 3-//4- as an electroactive probe. The concentration of the redox probe has been omitted, as well as the information on the supporting electrolyte. Large supporting electrolyte concentrations are necessary to decrease the solution resistance and limit analyte migration. Also, the presence of chloride ion will ensure the proper functioning of the pseudo-reference electrode if such was utilized in the experiment.

Author Response

In the presented manuscript, a novel aptasensor based on ordered mesoporous carbon/2D Ti3C2 MXene was fully characterized and successfully utilized in real samples analysis. The authors already presented a similar aptasensor for AAs detection in milk (please see ref. 19). The previous aptasensor was based on ordered mesoporous carbon only; however, it exhibited a similar performance. A comparison of those two aptasensors should be made, the differences underlined and demonstrate how the introduction of Ti3C2 MXene influenced its performance.

Response: We agree with you and thanks for your suggestion. The published paper (ref. 19) is a preliminary research result of our research group, and I know it very well. In the previous method, the aptasensor was based on ordered mesoporous carbon (OMC) and displayed a linear range from 10 nM to 1000 nM with limit of detection of 2.47 nM. However, in the assembly process of our previous aptasensor, we found the single OMC was easy to fall off the electrode surface because of the porous structure. In order to make the material better modified on the electrode surface and improve the detection performance of aptasensor, we further modified the OMC on Ti3C2 MXene and then prepared nanocomposites of OMC@Ti3C2 MXene. Under this condition, owing to the excellent layer structure and desirable biocompatibility of Ti3C2 MXene, the OMC@Ti3C2 MXene nanocomposites could be well modified on electrode surface. The aptasensor based on OMC@Ti3C2 MXene nanocomposites showed a wider linear range from 10 nM to 2000 nM, and the limit of detection slightly increased to 3.51 nM. Moreover, the stability of aptasensor was further improved from 14 days to 18 days. We have supplemented these in our revised manuscript.

Furthermore, some discrepancies occur in the text, especially in the ‘Material and Methods’ section, which may confuse the reader. Also, some details on the experimental setup are missing.

Response: We agree with you and thank you very much for reminding me. We have checked the whole text and supplemented the relevant information in our revised manuscript.

  1. The Authors stated that broad spectrum aptamer were synthesized. The plural form indicates that more than one sequence was tested.

Response: Thank you very much for reminding me. We are sorry for our mistakes. The number of aptamer sequence used in this manuscript is one, and we wrote the singular in the plural. We have corrected the content as follows:

The broad-spectrum aptamer for AAs was synthesized and purified from Sangon Biotech Co., Ltd (Shanghai, China) and the sequence was as follows: 5’-CGGATCCCCAGCTCGGGGTGCTATGGAGGCTGTATCGGAGACCTGCAGG-3’. (line 112-114).

  1. The SPCE electrode was utilized in the presented work. The type of counter and reference electrode should be specified as well.

Response: Thanks for your suggestion. The SPCE electrode used in our work was a three-electrode system with a carbon working electrode, a carbon counter electrode and an Ag/AgCl reference electrode. We have supplemented these in our revised manuscript.

  1. The liquid chromatography-tandem mass spectrometer was used for the quantitative analysis of AAs in milk samples. For clarity, it should be specified that triple quadrupole type MS was used. The ion source (ESI/ APCI/ AJS ?), operation mode (MRM mode could be used for quantitative analysis), column type, and elution program should also be indicated.

Response: We agree with you and thanks for your suggestion. The quantitative analysis of AAs in milk samples was carried out on the Agilent 6470 Liquid chromatography-tandem mass spectrometry (LC-MS/MS) equipped with triple quadrupole MS (Agilent Technolo-gies, USA). Electrospray positive ionization (ESI+) was used for ionization of the target analytes and multiple reaction monitoring (MRM) mode was used to determine the quantitative and qualitative ions. The samples were purified by C18 solid phase extraction column. The elution program was gradient elution with methanol and 0.1% formic acid aqueous solution.

  1. In the description of the Ti3C2 MXene preparation, at the final step, the supernatant was collected after centrifugation and stored before use. It is rather bizarre as the MXene is the precipitate.

Response: Thank you very much for reminding me. We are sorry for our mistakes in describing the method. At the final step, the black precipitate was collected after centrifugation and stored before use. We have corrected the mistake in our revised manuscript.

  1. The methodology for the calibration of the aptasensor is not clear. In the current state, it seems that the aptasensor was incubated in the mixture of 10 different AAs of a final concentration of 10, 55, or 100, etc. Was that an equimolar mixture? What was the sensor performance (sensitivity, LOD, LOQ, linear range) in the presence of only one antibiotic? 8 µL of the AAs solution was dropped on the aptasensor and left to dry. Then, the sensor was placed in Fe(CN)6 3-/4- solution, and the DVP signal was registered. How did this drying step influence the aptasensor response? Were any control experiments conducted? Various incubation times with AAs were evaluated. Was the drying step involved here as well?

Response: Thank you for your question. In our work, the aptasensor used the broad-spectrum aptamer as biorecognition element. In the presence of AAs, the broad-spectrum aptamer could simultaneously identify AAs in solution. Thus, the aptasensor was applied for simultaneous detection of AAs. And, the change in the DPV response of AAs containing KAN, NEO, TOB, DH-STR, GEN, SPE, STR, AMI, APR and PAR was detected, respectively, for verifying the detection performance of the aptasensor for each AAs. Then the aptasensor was incubated with the equimolar mixture of 10 different AAs of a final concentration of 0, 10, 50, 100, 250, 500, 750, 1000, 1500 and 2000 nM, and the calibration curves of mixed targets of the aptasensor was established, which was in accordance with recognition performance of broad-spectrum aptamer.

      After the electrode surface modification and incubation, we let the electrode dry in the air. In the process of our experiment, we used dry step including dry in the air, dry in the oven and dry by infrared light irradiation. The DPV response was not stable when the electrode was dried in the oven and by infrared light irradiation, considering that it might be due to the high temperature might affect the combination of aptamer and target. Therefore, the dry step we adopt was dry in the air. The incubation time between aptamer and AAs was investigated. We controlled the incubation time by adding a special cover to prevent the volatilization of the solution. After incubation, the electrode was dried in the air. The drying step is not within the incubation time.

  1. The DPV and EIS measurements were conducted using Fe(CN)6 3-//4- as an electroactive probe. The concentration of the redox probe has been omitted, as well as the information on the supporting electrolyte. Large supporting electrolyte concentrations are necessary to decrease the solution resistance and limit analyte migration. Also, the presence of chloride ion will ensure the proper functioning of the pseudo-reference electrode if such was utilized in the experiment.

Response: Thank you for your comment. In this work, the [Fe(CN)6]3−/4- solution contained 5.0 mM K4[Fe(CN)6], 5.0 mM K3[Fe(CN)6] and 0.1 M KCl, which was prepared with ultrapure water as solvent. We added chloride ion by KCl in the experiment to ensure the proper functioning of the pseudo-reference electrode. We have supplemented these in our revised manuscript.

Round 2

Reviewer 3 Report

Thank you for the comprehensive response to my points. In general, the corrections added to the text improve the quality of the manuscript. The explanation of the novelty of the conducted research yet leaves one unsatisfied.

The working parameters of the previously described and current aptasensor were compared. However, comparing LOD that was determined using a different approach (LOD = 3 σ/S vs. S/N = 3) might not give a clear picture. Also, it seems that higher concentrations of AAs were not tested in the case of the previously reported aptasensor. The statement that the current one exhibits a wider linear range might be a slight exaggeration. The Authors pointed out the drawbacks of the former aptasensor as ‘the single OMC was easy to fall off the electrode surface because of the porous structure in the assembly process of aptasensor’ (p. 2, lines 79-81). This statement is unclear to the reader and should be further elucidated.

The description of LC-ESI-MS measurements is still missing crucial information. It is stated that the MRM mode was used to determine the quantitative and qualitative ions. Which ions were analyzed? The question, in fact, should be, which transitions were monitored? Also, regarding the LC system, the gradient program should be included. Moreover, the statement ‘The samples were purified by C18 solid phase extraction column’ is imprecise. In an LC system that is hyphened to MS, the separation of the compounds occurs, not the purification/extraction.

The change in DPV current before and after incubation in AAs/ other antibiotics was used in AAs quantitative analysis, and to determine the selectivity of the developed sensors. Again, were any control experiments conducted for blank samples, i.e., where no antibiotics were present in the sample, to verify the impact of the determination procedure on the current response? The manuscript states that samples containing 0 nM AAs were analyzed, but no information on the outcome is included. Typically some variation in current response is observed for blank as well. This information, for instance, could be presented in Figure 6A.

Minor language corrections are still needed. Please go through the manuscript and ensure that correct verb forms and adjectives are used and that the sentences contain subject and verb (using a free Grammarly editor might be helpful). For example, in the Abstract and on page 2, lines 48- 50, the sentences appear to be missing a subject:

'In view of the class of antibiotics is able to be detected simultaneously, it will avoid analyzing numerous samples and simplify procedure of detection for food safety.'

'In view of the class of antibiotics is able to be detected simultaneously, it will avoid analyzing numerous samples and simplify procedure of detection so as to improve the detection efficiency'

Further, the sentence on page 12, lines 373-375, is confusing. Please clarify the applied methodology.

On page 3, line 134, the abbreviation SEM is defined as Scanning electron micrographs, which is incorrect. SEM stands for a scanning electron microscope.
